# Psychometric Properties of the Polish Version of the Sports Anxiety Scale-2 (SAS-2)

**DOI:** 10.3390/ijerph20146429

**Published:** 2023-07-22

**Authors:** Kanupriya Rawat, Aleksandra Błachnio, Krzysztof Suppan

**Affiliations:** Department of psychology, Kazimierz Wielki University, 85-064 Bydgoszcz, Poland; alblach@ukw.edu.pl (A.B.); krzysztof.suppan1@gmail.com (K.S.)

**Keywords:** cross-cultural adaptation, psychometrics, anxiety, athletes

## Abstract

The main aim of the study was to assess the psychometric properties of the Polish version of the Sports Anxiety scale-2 (SAS-2). The study covered 396 athletes, ages ranging from 18 years to 35 years. The task and ego orientation in sport questionnaire (TEOSQ) and State and Trait Anxiety Inventory (STAI) were used for validation of the scale. Cronbach’s alpha for the somatic anxiety subscale was 0.88, for the worry subscale was 0.95, and for the concentration disruption was 0.86, respectively. The three-factor model and hierarchical model fits perfectly as CFI > 0.95, TLI > 0.95, and RMSEA < 0.08. Satisfactory results in measurement invariance show the use of the tool for any gender or athlete (high-performance, recreational) group. The internal consistency (α = 0.86–0.95) and the test–retest reliability (ICC = 0.87–0.90) were satisfactory. There was a statistically significant negative correlation between task orientation and total anxiety along with its three subscales, and a weak positive correlation between ego orientation and worry subscale. Meanwhile, a weak to moderate positive correlation was exhibited between total anxiety and its subscales with the STAI-T and STAI-S. Additionally, female, recreational, and female recreational athletes’ groups had weak negative associations between worry and concentration disruption trait anxiety and task orientation, and a weak positive association between somatic anxiety and ego orientation.

## 1. Introduction

Sports is a context in which an athlete intensively develops himself, competes, strives to win, and improves his skills. This process of self-growth exposes him to experiencing anxiety. The importance of sports anxiety is documented by a large number of studies conducted [1,2,3]. Athletes frequently feel excessive anxiety when competing [4]. In addition, 63% of competitors feel anxious before a competition [5]; hence, the construct of pre-competitive anxiety has also been tested [6]. The multidimensional anxiety theory [7] asserts that competitive anxiety, including state and trait anxiety, can occur at the somatic and cognitive level. Competitive anxiety (CA) is an emotional response that frequently manifests before or during sporting events. The multidimensional construct of anxiety in sports is made up of two components: somatic and cognitive. According to Martens et al. [6], the two main causes of cognitive anxiety are “negative expectations about success or negative self-evaluation”. Cognitive dimensions include the content of thoughts, such as poor performance, unfavourable evaluation, social comparison, expectations, etc. [8,9], all of which can have a significant negative impact on an athlete’s performance. While many reports indicate the costs associated with experiencing sports anxiety, there are also fewer studies showing the benefits of anxiety in sports [10]. The physiological and affective aspects of anxiety that stem directly from autonomic arousal are referred to as somatic anxiety [6]. The somatic dimension includes a variety of physiological reactions, such as muscle tension, tremors, sweating, etc. Each component has a distinct relationship to athletic performance. 

The multidimensional anxiety theory has been the foundation for the development of two questionnaires. The Sport Anxiety Scale-2 (SAS-2) focuses on sport-specific trait anxiety, which was originally defined by Martens, while the Competitive State Anxiety Inventory-2 (CSAI-2; [7]) focuses on the situational occurrence of the phenomenon. The CSAI-2 [7] and CSAI-2R [11] are the most commonly used multidimensional scales with signs and symptoms for assessing competitive state anxiety experienced by athletes, whereas the SAS-2 [12] is the most commonly used scale for assessing competitive trait anxiety. Smith et al. [13] created the Sport Anxiety Scale (SAS) to measure how athletes’ cognitive and somatic anxiety differs from person to person and how their trait anxiety changes over time. It consists of 21 items that assess anxiety response tendencies in two aspects: somatic and cognitive. Based on exploratory and confirmatory factor analysis, cognitive anxiety is further subdivided into worry and concentration disruption [13,14]. However, Smith and his colleagues [12] found that because high school and college athletes were involved in the development of SAS items, some of the original SAS items may have been difficult for children to understand and respond to. Therefore, they revised SAS using exploratory factor analysis (EFA) and confirmatory factor analysis (CFA) to extract 15 items from 30 new or revised items, replicating the original SAS structure in samples of different ages. The 15 items in this updated version, named SAS-2, were evenly distributed across the three subscales (somatic anxiety: items 2, 6, 10, 12, and 14; worry: items 3, 5, 8, 9, and 11; and concentration disruption: items 1, 4, 7, 13, and 15) and were scored on a four-point scale of intensity (1 = not at all to 4 = very much). In analyzing the psychometric properties of this instrument, CFA indicated that the three-factor structure with or without a higher-order global anxiety component was the most appropriate [comparative fit index (CFI) = 0.95–0.97; root mean square error of approximation (RMSEA) = 0.05–0.065], and satisfactory values were also found for internal consistency (α ≥ 0.84) and test–retest reliability (≥0.76) [12].

SAS-2 adaptations in various languages or countries, including Spain [15], Brazil [16,17], Belgium [18], Portugal [19], Malaysia [20], Indonesia [21], and Korea [22], as well as numerous validity and reliability studies, have confirmed the scale’s high diagnostic value for psychological assessment in sports, hence making this scale popular worldwide. The validity of the three-factor model was confirmed in most adaptations. For example, in a study by Ramis et al., [23] analyses were conducted on samples recruited in Spain [15], Belgium [18], and Portugal [19]. The results confirmed the three-factor model for all of these versions of SAS-2 with CFI and TLI above 0.92 and RMSEA = 0.04. Similarly, the Malaysian version also confirmed the three-factor structure of SAS-2, whereas the Korean version of SAS-2 confirmed the three-factor model and the three-factor higher order model with both having good fit indices with CFI = 0.92, RMSEA = 0.07, and SRMR = 0.06. In the Brazilian sample, the original SAS-2 model (i.e., three-factor model) was not a good fit. However, the improved version of the model included a correlation between errors of item 6 and item 12 on the somatic anxiety subscale where CFI = 0.97, TLI = 0.96, and RMSEA = 0.08. Similarly, in the Indonesian version, the original three-factor model had good fit indices in terms of RMSEA = 0.77, CFI = 0.92, TLI = 0.91, and PNFI = 0.73 but according to chi-square, the three-factor model was not a good fit which could be improved by using large samples. 

To make personalized interventions for athletes, research on competitive anxiety has focused on finding out how anxiety symptoms vary by gender, age, type of sport, and other psychological factors. With regard to gender, female athletes typically reported higher levels of precompetitive state anxiety, factors associated with worries, and global competitive trait anxiety [7,24]. Correia and Rosado [1] reported females having higher levels of general anxiety, somatic anxiety, and concentration disruption, while Ramis et al. [23] reported slightly higher levels of anxiety worry in women than men. Additionally, some researchers contend that gender mediates the relationship between the causes and effects of anxiety. Grossbard et al. [25] found that the motivational climate and anxiety, as well as anxiety and performance, had different effects on each other.

In Poland, the cross-cultural adaptation of SAS-2 was conducted independently by two research teams: Tomczak et al. [26] and ours. Both projects were conducted at the same time, although Tomczak et al. [26] succeeded in finishing it early. They published their results in an article, through which we learned about their work. As a result, we compared the two versions of SAS-2 in Polish and noticed minor differences in the way statements were addressed. These changes were discussed with linguists and sports psychologists, who all agreed that the meanings were the same. After discussing the situation, we decided to complete our project, seeing its replication value. Our study had additional value; it tested the criterion validity of SAS-2, which was lacking in the previously published adaptation [26]. We used an established anxiety assessment scale (STAI) as the criterion measure and expand the relationship of SAS-2 and STAI through this study. Furthermore, our models provided a better fit to the data (CFI = 0.969, TLI = 0.962, RMSEA = 0.061, SRMR = 0.036) than the models used in the other Polish study i.e., the three-factor model: CFI = 0.945, TLI = 0.933, RMSEA = 0.072, SRMR = 0.046. Further the model fits for male–female, high-performance, and recreational athletes were also better. Our reliability scores (0.86–0.95) were also higher than the previous Polish version of SAS-2 (0.81–0.91). Additionally, we tried to emphasize the issue of identical fit indices in the hierarchical and three-factor model, which was seen in some adaptations [22,26] including our own study. 

Overall, this study aims to investigate the psychometric properties of construct validity and reliability of SAS-2 in a Polish context. It also aims to fill a gap in previous research by testing criterion validity and expanding research on relationships between the SAS-2 and other scales, such as STAI and TEOSQ, which will broaden research on the scale’s validity.

## 2. Materials and Methods

### 2.1. Sample

This study comprised 396 Polish athletes, (male = 205, female = 191; high-performance athletes = 210; recreational athletes = 186); aged from 18 to 35 years old (M = 22.89, SD = 3.82) who participate in various sports, including football, hockey, volleyball, handball, gymnastic, athletics, judo, MMA, karate, tennis, table tennis, badminton, and cycling. High-professional athletes were either playing for professional sports clubs or national teams, while recreational athletes reported playing in sports clubs or the university’s sports club, and they also reported their participation in sports competitions from time to time. 

### 2.2. Measures

#### 2.2.1. Demographics

Participants responded to demographic questions (e.g., gender, age, domicile, sport types, years of training, frequency of training, and level of sports participation). In addition to the Polish version of the Sports Anxiety Scale-2, the study employed the following instruments to assess convergent, divergent, and discriminant validity.

#### 2.2.2. Sports Anxiety Scale-2

A Polish translation of the SAS-2 was used in this study. The adaptation procedure is given below. The original SAS-2 scale was developed in the English language by Smith et al. [12], which consists of three subscales, somatic anxiety, worry, and concentration disruption. The scale comprises 15 items rated on a four-point scale, ranging from 1 (not at all)—to 4 (very much). The sum of the items from each subscale yields the score, which ranges from 5 to 20 points and indicates how likely it is that someone will experience competitive anxiety.

#### 2.2.3. Adaptation Procedure

The SAS-2 was translated into the Polish language by three linguistic experts. The translation was compared and reviewed by authors and a sports psychologist. The final version was then back-translated by two other psychologists who were fluent in English. The translations were reviewed by the authors and discussions were held for minor changes. The final version of the scale was checked for the readability of items and instructions, along with an analysis of factor structure among a sample of (N = 150) participants. Later, a sample of (N = 246) participants completed the anonymous paper-and-pencil mode of the Polish version of SAS-2 along with measures used for validation purposes. 

To select the sample, the following inclusion criteria were used: recreational and high-professional athletes of any of a variety of sports modalities and either sex, age 18–35 years and participation in sports competitions at least once per month. This study was approved by the ethical committee of Kazimierz Wielki University. All subjects voluntarily participated in the study and signed the informed consent form.

#### 2.2.4. Task and Ego Orientation in Sport Questionnaire (TEOSQ)

We used the Polish version of this scale, adapted by Tomczak et al. [27]. It consists of 13 items where task orientation comprises seven items and ego orientation comprises six items. Items were rated on a scale of 1 (strongly disagree) to 5 (strongly agree). The Polish version of the TEOSQ has adequate internal consistency (α = 0.84 for the ego subscale, and 0.81 for the task subscale). 

#### 2.2.5. State–Trait Anxiety Inventory (STAI)

State and trait anxiety was measured with STAI developed by Spielberger et al. [28]. The Polish adaptation of STAI used in this study was adapted by Wrzésniewski et al. [29]. It consists of 40 items which are answered on a self-reporting basis. STAI measures two types of anxiety: state anxiety (anxiety about an event), and trait anxiety (anxiety as a personal characteristic). Each subscale includes 20 items rated on a 4-point Likert scale: 1—not at all, 2—somewhat, 3—moderately, and 4—very much for Form Y1 and 1—almost never, 2—sometimes, 3—often, and 4—almost always. 

### 2.3. Statistical Analysis

TIBCO Statistica and R version 4.2.2 were used for analyzing this study. Due to the significant multivariate non-normality of the data, analyses were performed using the robust maximum likelihood estimator (MLR). The first stage of the analysis consists of construct validity which was evaluated using confirmatory factor analysis. Based on theories, a hierarchical model with total anxiety and its three subscales: somatic anxiety, worry, and attention disruption was estimated. The 3-factor model with only three subscales without total anxiety was then estimated. Then, the 1-factor model (without extracting the subscales of the tool) was estimated and compared with the other two models, to assess the validity of the tool. The postulated models were assumed to have a satisfactory fit to the data if CFI and TLI were greater than 0.90, and RMSEA and SRMR were less than 0.08 [30]. The Satorra–Bentler correction was used because the distribution’s multivariate normality was not maintained [31].

Next, the measurement invariance of SAS-2 was examined across gender and sports participation level (high performance vs. recreational). Configural, metric, scalar, and strict invariance was estimated. A significant difference between groups will be seen if there is a decline in CFI below 0.01, a rise in RMSEA above 0.015, an increase of 0.01 for strict and scalar invariance, and 0.03 for metric invariance for SRMR [32].

Item reliability was evaluated by looking at the values of factor loadings and their squared values (the proportion of variance in an item due to a given construct), which led to the analysis of convergent validity indices of the SAS-2 scale. For minimum standardized loading values, criteria ranging from 0.40 (the most flexible) to 0.70 (nearly 50% of the variance in the item explained by the factor) are commonly used. The average extracted variance (AVE) and composite reliability (CR) values for each subscale were then determined [33].

When AVE is less than 0.5 but the CR requirement is met, we can say that the construct satisfied the convergent validity requirement. On the other hand, discriminant validity was assessed by comparing the maximum shared variance (MSV), which is the highest value of the square of the correlation between a given latent variable and another construct-latent variable, with the AVE of a particular latent variable. The discriminant validity of the construct is proven when the AVE value surpasses the MSV [33].

Additionally, Cronbach’s alpha (N = 396) as well as the interclass correlations (ICC; N = 100) were used to investigate the reliability of the scale. The discriminatory power of a particular test item was calculated by its correlation with the total subscale score.

A Mann–Whitney U-test was performed to compare participants’ mean anxiety and subscale scores across gender and their level of sport participation. In order to examine the relationship between anxiety and goal orientation and state or trait anxiety, the Spearman correlation method was applied to a sample of (N = 246) participants.

## 3. Results

### 3.1. Characteristics of the Sample

The descriptive statistics results (Table 1) show that the worry subscale had the highest average value (10.33 ± 4.04), followed by somatic anxiety (7.98 ± 2.76) and concentration disruption (7.63 ± 2.64). Overall, the mean score of total anxiety was (25.94 ± 8.07).

### 3.2. Factor Validity and Reliability Estimates

The overall fit test results of the CFA models are presented in Table 2. They showed a satisfactory fit to the data (i.e., CFI, and TLI > 0.90, RMSEA, and SRMR < 0.08) for all models except the 1-factor model. Both hierarchical and 3-factor models (Table 2) fit the empirical data well. Additionally, the 3-factor model has satisfactory fits to the data in terms of gender and level of sports participation groups (Table 2).

Table 3 shows factor loadings and reliability estimates, as well as squared multiple correlations, CR, AVE, and MSV. All factor loadings were significant (*p* < 0.001) and ranged from 0.685 to 0.94. We assumed discriminant validity since the average variance explained (AVE) was more than the maximum shared variance (MSV) for each of the subscales, with the exception of the concentration disruption scale, where the AVE value of 0.303 was just slightly higher than the MSV value of 0.30.

The Cronbach alpha coefficient for the whole scale was 0.93. For all subscales of SAS-2, Cronbach’s alpha coefficients ranged from 0.86 to 0.95 (Table 3) which indicates sufficient internal consistency. The discriminant power coefficients for the items of somatic anxiety, worry, and concentration disruption subscales were, respectively: 2-0.80, 6-0.844, 10-0.79, 12-0.80, and 14-0.84; 3-0.90, 5-0.87, 8-0.91, 9-0.93, and 11-0.89; 1-0.82, 4-0.80, 7-0.83, 13-0.77, and 15-0.74. Interclass correlations (ICC) for Polish version of SAS-2 subscales were: Total anxiety = 0.938, somatic anxiety = 0.891, worry = 0.871, and concentration disruption = 0.902. Overall, the high ICC values suggest good reliability or consistency of the measurement over the two-week interval.

### 3.3. Measurement Invariance

The results of invariance analysis for the Polish version of SAS-2 are shown in Table 4. The postulated models were analysed for gender and level of participation groups and they showed acceptable fit to the data. Therefore, the Polish version of SAS-2 scale can be relied upon as a universally valid and reliable instrument, regardless of respondent’s gender or level of sport participation as the following situation was not observed in any case (i.e., the decrease in CFI greater than 0.01 and the increase in RMSEA greater than 0.015).

### 3.4. Assessment of Competitive Anxiety Based on Gender and Level of Sports Participation

Looking at the differences between gender, the data were almost identical; therefore, no significant differences were found. Only for somatic anxiety, a statistically significant effect of gender was found (Table 5). Men had lower mean anxiety levels than women. In addition, no significant differences were found between the levels of sports participation (Table 5).

### 3.5. Correlations between Competitive Anxiety, State and Trait Anxiety, and Goal Orientation

Through Spearman correlation, statistically significant, negative weak associations were found between anxiety (General and subscales: somatic anxiety and concentration disruption) and task orientation (TEOSQ subscale) in the whole group of the examined athletes. Correlations between anxiety and goal orientation were reported in Table 6. Similarly, a medium-strong association was found between total anxiety, its subscales and state (STAI X-1), and trait anxiety (STAI X-2).

## 4. Discussion

Utilizing confirmatory factor analysis, this study modified and examined the precision of the Sport Anxiety Scale-2 (SAS-2) instrument in Polish. CFA supported the hierarchical and the three-factor model. Performance anxiety is viewed as a global construct with three associated somatic and cognitive subcomponents in the theoretical model behind the SAS [12]. SAS-2 is designed to assess the total score of anxiety, along with the scores of three subscales. Researchers can use total anxiety score, subscales scores, or both. The hierarchical model was evaluated as some researchers prefer to use the total score as a general index of sports performance anxiety along with its subscales. It is valuable in case we need to know the overall anxiety score of the athlete along with its subscales. However, three-factor model is the sum of the scores of three subscales separately. It can be useful to evaluate the athlete’s anxiety in terms of somatic and cognitive aspects. It can be useful for the coach or trainers to identify which aspect they need to focus on for the better performance of the athlete. When comparing the obtained results to those of other authors, it can be said that the adapted scale’s three-factor and hierarchical models, which were both estimated on the total sample, provided better fits to the data than the models used in the Polish [26], Spanish [15], Indonesian [21], and Korean [22] versions of the scale. Our model is similar in the good fit to the data to the Brazilian [17] and Original versions of the scale [12]. The three-factor model fits the empirical data well for the total sample and in groups (gender and level of sports participation). Additionally, the three-factor model was better fitted than the previous study, with CFI > 0.95, TLI > 0.95, and RMSEA < 0.08. [26]. In line with earlier research findings, high correlations were also identified between the factors corresponding to each of the anxiety subscales. An issue that still needs to be resolved concerns the identical fit indices for the hierarchical and three-factor model obtained in some studies [22,26]. Such values were also obtained in the current analyses. The question of a better fit between the hierarchical model and the three-factor model needs to be resolved. Currently, in the literature, the hierarchical model along with the three-factor model is mentioned in the original version of the scale [12], as well as the Korean [22] and Polish ones [26], and in our study, while other adaptations only mentioned the three-factor model in their study [17,21,23]. Moreover, only the original study [12] has different degree of freedom and fit indicators for the hierarchical and three-factor models (df = 87 for three-factor model, and df = 89 for higher order model). We believe that this issue emerged due to the small sample size of our study. The difference between the hierarchical and three-factor models in terms of degree of freedom and fit indicators is only seen in the original study of SAS-2 [12] which has the largest sample size compared to other adaptation studies. If the sample size is relatively small, it may limit the ability to detect meaningful differences between the models. With a larger sample, the added complexity of the hierarchical model might be better detected and result in different fit indicators. However, until now only the Korean [22], Polish [26], and our study have identical fit indices and we all have sample sizes smaller than the original study, i.e., 850 (used for factorial validation using CFA) [12]. Therefore, small sample size seems to be the best reasoning to explain the identical fit indicators of hierarchical and three-factor model.

Factor loading values of all items in this Polish adaptation were fairly high (≥0.50), which shows that all the items were acceptable [34,35,36]. The smallest and largest factor loading values were 0.685 and 0.94, respectively (see Table 3). The AVE values for somatic and concentration disruption were 0.35 and 0.3, respectively, which is below the threshold value (i.e., <0.50); however, AVE for the worry subscale was satisfactory. Low AVE values could be caused by factor loadings ≤0.70 [34]. This result is in line with the study carried out by Cho et al. [22]. Low AVE values were expected for concentration disruption in previous studies; however, for somatic anxiety few studies support the low values of AVE [21]. Our results are in line with these facts. Convergent validity is satisfactory as the somatic anxiety and worry subscale showed high composite reliability values, 0.727, and 0.825, respectively. For concentration disruption, the CR value is 0.683 which is slightly below the threshold value CR < 0.7; however, according to Hair et al. [36], a CR value of ≥0.7 is good reliability, while between 0.6 and 0.7 is acceptable reliability. Considering this claim, we can say that the three Polish SAS-2 subscales (somatic anxiety, worry, and concentration disruption) have satisfactory internal consistency reliability. In addition, AVE for every subscale was higher than MSV, allowing us to assume the discriminant validity of the tested constructs.

This Polish version of SAS-2 has a high level of internal consistency. Cronbach’s alpha coefficient for each subscale ranges from 0.86 to 0.95. These findings are consistent with the reliability test of the original version [12] and the Brazilian version [17] with a Cronbach’s alpha coefficient value of ≥0.80. Our reliability scores are also higher than the previous Polish version of SAS-2, 0.81–0.91 [26].

Additionally, it was demonstrated that the measurement was independent of sport participation levels (high-performance, recreational) and gender (men, women). In both cases, one of the four basic types of measurement invariance was shown: configural, metric, scalar, or strict (see Table 4). Every group evaluated (i.e., gender: male, female; level of sports participation: high-performance, recreational athletes) is assumed to use the scale in a similar manner. The results of our study support the measurement invariance in high-performance and recreational athletes studied by a previous study on the Polish version of SAS-2 [26] and previous studies on measurement invariance by gender, e.g., [23,37,38].

The average level of anxiety by gender and level of sport participation were also compared in this Polish SAS-2 scale validation study. According to research, general anxiety and its components are more prevalent among women than men [26]. We obtained the same outcomes (see Table 1). The observed difference can be attributed to women’s higher levels of general emotionality and sensitivity, which are linked to higher levels of anxiety than men. Males also reported higher levels of self-confidence than females, which may have helped them in some way to guard against harmful symptom interpretations [39]. This could be one explanation for these differences. As an alternative, it has been asserted that females are more willing than males to report their feelings, especially those that are unpleasant to Competitive Anxiety in Sports in nature [40]. Therefore, because they perceive greater social acceptability of reporting anxiety, women may be more likely to present a more accurate reflection of their symptoms. 

Our findings largely agree with those of studies conducted with participants aged 18 or older, such as those by Tomczak et al. [26] and Martinez-Moreno et al. [41]. Additionally, no differences were discovered between high-performance and recreational athletes in our study regarding the intensity of anxiety or any of its components (see Table 5). However, one might expect that high-performance athletes are likely to experience lower levels of anxiety due to emotional resilience. These effects may be the opposite, resulting in similar levels of anxiety in the compared groups due to the concerns of high-performance athletes regarding their sporting results, which are subjected to personal and social expectations and social evaluation. These elements might make anxiety even more intense. Given that competitive settings are complicated and situational elements such as competition can occasionally change how propositional goal orientations affect performance, these effects may be somewhat opposing given the identical levels of anxiety in the comparison groups.

In our attempt to find a correlation between competitive anxiety and task and ego orientation (see Table 6), there was a weak positive association between competitive anxiety (total anxiety and worry subscale) and ego orientation of TEOSQ which were slightly similar to previous studies [12,26]. As the association is found only in the total anxiety and somatic anxiety subscale, especially in female, recreational, and female-recreational athlete groups. However, we also discovered that the same group (female, recreational, and female-recreational athletes) (see Table 7) was the most susceptible to the weak negative relationships between worry and concentration disruption trait anxiety and task orientation (as indicated by the TEOSQ questionnaire). As a result, the context of this relationship may be critical. For instance, the connection appears to be strongest among women (especially recreational athletes), who typically engage in less intense competition. There may be more latitude in how the situation is viewed and how it is handled when people play sports for recreation. Additionally, one’s expectations of the outcomes may change in this situation. As a result, there may be significant differences in how women who engage in recreational sports perceive and interpret events, with women who have high expectations for the outcome of the competition acting more aggressively (ego-oriented), they feel more anxious (especially somatic anxiety). Meanwhile, female athletes or female recreational athletes who focus on their progress and development tend to experience lower levels of cognitive trait anxiety. This scenario is supported by previous studies [42,43,44]. However, more studies will be needed to fully understand the findings in these areas.

Regarding SAS-2 subscales’ correlation with STAI, it shows a weak to moderate correlation for STAI-T and moderate correlation for STAI-S which suggests that to some extent these two measures are measuring the same underlying construct. Previous studies support these findings to an extent, as some studies had a strong correlation with both state and trait anxiety of STAI with SAS-2 [45] while other studies have shown a positive correlation of SAS-2 with trait anxiety of STAI [17]. Our result matches with the former results where SAS-2 has a positive correlation: moderate correlation with state anxiety and weak to moderate correlation with trait anxiety. However, more studies should be conducted to find a stronger correlation with STAI-T as both of these scales measure trait anxiety. 

Although the study is in line with previous research and shows good psychometric properties, it has several limitations, as the sample size was not that large, and dividing them into groups (according to gender and level of sports participation), further made the groups smaller, which could possibly lead to identical fit indices in hierarchical and three-factor models. To add to the enrichment and reliability of the data, future research may also make use of more sophisticated diagnostic tools for psychiatric disorders, which was previously attempted by the Brazilian adaptation [17]. It would also improve the validity of the scale to examine anxiety variations between individual and team athletes or according to age groups.

This study aimed to prove the reliability and validity of the Sports Anxiety Scale-2 in the Polish population and to overcome the limitations suggested by the previous Polish adaptation of SAS-2. We were successful in having similar, rather better results, than the study by Tomczak et al. [26] and also addressed the limitations raised by their study regarding the expansion of research on relationships between the SAS-2 and other scales such as STAI, which also helped in ensuring the convergent validity of SAS-2. In addition, we pointed out an issue regarding identical fit indices in hierarchical and three-factor model which could be possibly due to small sample size. Overall, it displays the strong psychometric properties of Sports Anxiety Scale -2 in Polish context. As a result, when applied to athletes, the psychometric characteristics of SAS-2 can be used to measure trait anxiety (trait somatic anxiety, trait worry, and trait concentration disruption).

## 5. Conclusions

This study tested the accuracy of the Polish version of the Sport Anxiety Scale-2 (SAS-2) instrument model using confirmatory factor analysis. The Polish version of SAS-2 has a high level of internal consistency and convergent and divergent validity. A statistically significant, negative correlation between task orientation and total anxiety along with its three subscales and a weak positive correlation between the ego orientation and worry subscale was obtained. In addition, the SAS-2 subscales and total anxiety showed a weak to moderate correlation with the STAI-T and STAI-S, and the measurement was independent of sport participation levels and gender. Competitive anxiety, task and ego orientation, and state–trait anxiety are all correlated; the female, recreational, and female-recreational athletes’ group had weak negative associations between worry and concentration disruption trait anxiety and task orientation, and weak positive associations between somatic anxiety and ego orientation. This study aimed to prove the reliability and validity of the Sports Anxiety Scale-2 in the Polish population but had a few limitations. Future research may use more sophisticated diagnostic tools and improve the validity of the scale.

## Figures and Tables

**Table 1 ijerph-20-06429-t001:** Mean and standard deviations of the Polish version of the SAS-2.

Sex Level of Participation	N	Somatic M (SD)	Worry M (SD)	Concentration Disruption M (SD)	Total Anxiety M (SD)
Total	396	7.98 (2.76)	10.33 (4.04)	7.63 (2.64)	25.94 (8.07)
Female	191	8.27 (2.79)	10.69 (4.16)	7.79 (2.84)	26.74 (8.36)
Male	205	7.72 (2.71)	9.99 (3.91)	7.47 (2.44)	25.19 (7.73)
Recreation	186	8.26 (3.15)	10.67 (4.53)	7.74 (2.85)	26.67 (9.08)
High performance	210	7.74 (2.33)	10.02 (3.54)	7.52 (2.45)	25.29 (7.02)
Female-recreation	90	8.44 (3.25)	10.9 (4.64)	7.79 (3.1)	27.13 (9.45)
Female-High performance	101	8.11 (2.31)	10.5 (3.7)	7.79 (2.6)	26.4 (7.29)
Male-recreation	96	8.09 (3.07)	10.45 (4.44)	7.7 (2.6)	26.24 (8.74)
Male-High performance	109	7.39 (2.31)	9.59 (3.33)	7.28 (2.29)	24.26 (6.62)

**Table 2 ijerph-20-06429-t002:** Fit test results of CFA models for the Polish version of SAS-2.

Models	N	Chi-Square (df)	*p*	CFI	TLI	RMSEA 90% CI	SRMR
1 factor	396	995.949 (90)	0.000	0.742	0.699	0.173 [0.163, 0.182]	0.117
Hierarchical	396	196.403 (87)	0.000	0.969	0.962	0.061 [0.050, 0.073]	0.036
3-factor	396	196.403 (87)	0.000	0.969	0.962	0.061 [0.050, 0.073]	0.036
M (3-factor)	205	174.402 (87)	0.000	0.95	0.94	0.076 [0.060, 0.093]	0.045
F (3-factor)	191	134.796 (87)	0.001	0.974	0.968	0.057 [0.037, 0.075]	0.045
HP (3-factor)	210	138.617 (87)	0.000	0.969	0.963	0.056 [0.038, 0.073]	0.041
R (3-factor)	186	165.537 (87)	0.000	0.958	0.949	0.076 [0.058, 0.093]	0.045

M: Male, F: Female, HP: High-Performance athletes, R: Recreational athletes, CFI: Comparative Fit Index, TLI: Tucker–Lewis Fit Index, RMSEA: Root Mean Square Error of Approximation, SRMR: Standardized Root Mean Square Residual—robust values based on Satorra–Bentler correction.

**Table 3 ijerph-20-06429-t003:** Reliability test results.

Dimensions	Items	λ	λ^2^	CR	AVE	MSV	α
Somatic	C2	0.755	0.570	0.727	0.35	0.33	0.88
	C6	0.784	0.615				
	C10	0.719	0.517				
	C12	0.743	0.552				
	C14	0.83	0.689				
Worry	C3	0.88	0.774	0.885	0.608	0.37	0.95
	C5	0.815	0.664				
	C8	0.918	0.843				
	C9	0.94	0.884				
	C11	0.844	0.712				
Concentration Disruption	C1	0.76	0.578	0.683	0.303	0.30	0.86
	C4	0.761	0.579				
	C7	0.77	0.593				
	C13	0.723	0.523				
	C15	0.685	0.469				

λ: Factor loading, λ^2^: The squares of the factor loading, CR: Construct Reliability, AVE: Average Variance Extracted, MSV: Maximum Shared Variance, α: Cronbach’s Alpha.

**Table 4 ijerph-20-06429-t004:** Measurement invariance of the Sport Anxiety Scale-2 invariance across gender and level of participation.

	CFI	RMSEA	SRMR	ΔCFI	ΔRMSEA	ΔSRMR
Gender (males vs. females)						
Configural	1.000	0.024	0.040	-	-	-
Metric	1.000	0.027	0.049	0	0.003	0.009
Scalar	1.000	0.024	0.049	0	−0.002	0
Strict	1.000	0.023	0.051	0	−0.001	0.002
Level of participation in sports (high-performance vs. recreational)						
Configural	0.999	0.016	0.036	-	-	-
Metric	0.999	0.012	0.042	0	−0.004	0.006
Scalar	0.999	0.012	0.043	0	−0.001	0.001
Strict	1.000	0.009	0.045	0	−0.003	0.002

CFI: Comparative Fit Index; RMSEA: Root Mean Square Error of Approximation, SRMR: Standardized Root Mean Square Residual—robust values based on Satorra–Bentler correction.

**Table 5 ijerph-20-06429-t005:** Descriptive data and significant differences in dimensions per comparison of gender and level of sports participation.

	Gender					Level of Sports Participation				
	M (N = 205) M (SD)	F (N = 191) M(SD)	Z	*p*	Effect Size	HP (N = 210) M (SD)	R (N = 186) M (SD)	Z	*p*	Effect Size
Somatic	7.72 (2.71)	8.27 (2.79)	−2.33	0.021 *	0.011	7.74 (2.33)	8.26 (3.15)	−0.94	0.354	0.004
Worry	9.99 (3.91)	10.69 (4.16)	−1.56	0.121	0.007	10.02 (3.54)	10.67 (4.53)	−0.96	0.342	0.004
Concentration disruption	7.47 (2.44)	7.79 (2.84)	−0.7	0.491	0.003	8.26 (3.15)	7.74 (2.85)	−0.24	0.813	0.001

M: Male, F: Female, HP: High-Performance athletes, R: Recreational athletes, Z, *p* = significance, * *p* < 0.05.

**Table 6 ijerph-20-06429-t006:** Correlations between competitive anxiety, goal orientation, and state and trait anxiety.

SAS-2 (N = 246)	TEOSQ- Ego Orientation	TEOSQ- Task Orientation	STAI-S (Form X-1)	STAI-T (Form X-2)
Total anxiety	0.16 *	−0.211 *	0.412 *	0.379 *
Somatic	0.049	−0.212 *	0.381 *	0.366 *
Worry	0.208 *	−0.133 *	0.327 *	0.292 *
Concentration disruption	0.116	−0.211 *	0.343 *	0.311 *

SAS-2: Sports Anxiety Scale-2, TEOSQ: Task and Ego orientation in sports questionnaire, STAI-S: State and trait anxiety inventory—State anxiety, STAI-T: State and trait anxiety inventory—Trait anxiety, * *p* < 0.05.

**Table 7 ijerph-20-06429-t007:** Correlations between competitive anxiety, state–trait anxiety, and goal orientation across gender, and level of sports participation.

SAS-2 (N = 246)	Male (N = 125)				Female (N = 121)			
	S	W	CD	TA	S	W	CD	TA
TEOSQ- Ego orientation	0.134	−0.002	0.111	0.105	0.291 *	0.104	0.127	0.221 *
TEOSQ- Task orientation	−0.128	−0.186 *	−0.132	−0.176 *	−0.163	−0.278 *	−0.30 *	−0.282 *
STAI- S (Form X-1)	0.32 *	0.346 *	0.333 *	0.397 *	0.32 *	0.399 *	0.345 *	0.413 *
STAI-T (Form X-2)	0.372 *	0.372 *	0.352 *	0.44 *	0.185 *	0.329 *	0.262 *	0.295 *
**SAS-2** **(N = 246)**	**High Performance (N = 130)**				**Recreational (N = 116)**			
	**S**	**W**	**CD**	**TA**	**S**	**W**	**CD**	**TA**
TEOSQ- Ego orientation	0.119	0.01	0.127	0.109	0.267 *	0.067	0.104	0.193 *
TEOSQ- Task orientation	−0.255 *	−0.172	−0.255 *	−0.279 *	−0.031	−0.254 *	−0.17	−0.16
STAI- S (Form X-1)	0.439 *	0.381 *	0.431 *	0.507 *	0.237 *	0.362 *	0.266 *	0.333 *
STAI-T (Form X-2)	0.352 *	0.385 *	0.365 *	0.44 *	0.228 *	0.326 *	0.254 *	0.312 *
**SAS-2** **(N = 246)**	**Female High Performance (N = 62)**				**Female Recreational (N = 59)**			
	**S**	**W**	**CD**	**TA**	**S**	**W**	**CD**	**TA**
TEOSQ- Ego orientation	0.207	0.123	0.144	0.198	0.348 *	0.113	0.117	0.245
TEOSQ- Task orientation	−0.205	−0.209	−0.322 *	−0.294 *	−0.136	−0.335 *	−0.297 *	−0.28 *
STAI-S (Form X-1)	0.438 *	0.426 *	0.37 *	0.502 *	0.253	0.391 *	0.33 *	0.366 *
STAI-T (Form X-2)	0.211	0.357 *	0.205	0.304 *	0.169	0.310 *	0.301 *	0.288 *
**SAS-2** **(N = 246)**	**Male High Performance (N = 68)**				**Male Recreational (N = 57)**			
	**S**	**W**	**CD**	**TA**	**S**	**W**	**CD**	**TA**
TEOSQ- Ego orientation	−0.0002	−0.15	0.077	−0.026	0.183	0.042	0.097	0.144
TEOSQ- Task orientation	−0.349 *	−0.189	−0.238	−0.334 *	0.079	−0.187	−0.003	−0.025
STAI-S (Form X-1)	0.38 *	0.268 *	0.452 *	0.452 *	0.22	0.338 *	0.184	0.298 *
STAI-T (Form X-2)	0.393	0.322 *	0.474 *	0.486 *	0.294	0.342 *	0.187	0.341 *

SAS-2: Sports Anxiety Scale-2, TEOSQ: Task and Ego orientation in sports questionnaire, STAI-S: State and trait anxiety inventory—State anxiety, STAI-T: State and trait anxiety inventory—Trait anxiety, * *p* < 0.05.

## Data Availability

The data generated and/or analyzed during the current study are available from the corresponding author upon reasonable request.

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
