# Peer review of "Psychometric Properties of the Polish Version of the Sports Anxiety Scale-2 (SAS-2)"

_ijerph, 2023, doi:10.3390/ijerph20146429_

Round 1
Reviewer 1 Report
Overall, this paper is a well-written and informative contribution to the study of the psychometric properties of the Polish version of the Sports Anxiety Scale-2 (SAS-2). The authors have provided a clear and concise overview of the SAS-2. The aims of the study are well-formulated, and the study design and data analysis appear to be rigorous and appropriate. This paper makes an important contribution to the literature by showing that SAS-2 is a useful measurement tool in other cultures.
I think that in the introduction, it would be useful to write a few sentences about the results of the factor analysis of SAS-2 in different countries. In addition to this, it would be useful to write about the practical aspects of the hierarchical model in the discussion, e.g., the importance and interpretation of the distinction between main scales and subscales.
Although the study was very carefully done, I nevertheless see a problem with the result of the CFA factor analysis. For the hierarchical and 3-factor models, the fit indicators are perfectly identical, which is not possible. In the hierarchical model, the degree of freedom should also be higher than 87 since there is another latent variable (a higher-order factor). The logic is not good in the specification either, since both models give good fit indicators, and the male-female, high-performance athlete, and recreational athlete models are designed for the 3-factor model. And from the introduction, it seemed that the higher-order model was considered to be the basic one. This is an important passage here, as the 3-factor model supports the use of 3 scales, whereas the hierarchical model supports the use of 3 subscales and 1 main scale (total score). This definitely needs to be clarified. In addition to this, it would also be useful to test the bifactor model, since then one could even calculate model-based reliabilities (hierarchical omega) to clarify how reliable the subscales and the main scale are separately (see, e.g., Reise 2012; Rodriguez et al., 2016). Together, this information and the theoretical context will enable the selection of the most appropriate factor model and ensure the validity of inferences drawn from scale scores.
I was pleased to read your well structured and precisely written manuscript. In my opinion, the additional notes and analysis mentioned above would greatly enhance the value of the paper.
References
Reise, S. P. (2012). The rediscovery of bifactor measurement models. Multivariate behavioral research, 47(5), 667–696
Rodriguez, A., Reise, S. P., & Haviland, M. G. (2016). Evaluating bifactor models: calculating and interpreting statistical indices. Psychological Methods, 21(2), 137–150.
Reviewer 2 Report
The manuscript meets the requierments of scientific research.
In the introduction it needs further explanation why two reseaech groups validated the same questionnaire at the same time, why the present validation procedure needs to be presented, how does it differ from the previously publised one?
The reseaerch process is well described, and consistently implemented. The methodology is correct, the results are clearly presented. One of the results of the article, that female recreational athletes had association between anxiety and task/ego orientation. A shotcoming of the research is that it does not report age-disaggregated findings. It can be assumed, that there are yunior athletes in the sample, whose anxiety may be related to the expected development of their sporting careers, which may have a significant impact of the results. The sports played by the athletes are also not mentioned in the article, although wether the athletes play individual or team sports, which can be infuencing factor aswell.
The answers to these questions need to be added to the manuscript.
Minor editing of English language is required.
Reviewer 3 Report
Thank You for giving me the chance to read tis interesting paper. I found it is important (specially for psychologists) to explore athletes emotions, and Your work helps and improves this task. I have only minor suggestions:
1. Why didn;t You tried exploratory factor analysis? It probably is a very interesting to explore the structure of measurement in Polish population not only test if the original is suitable,
2. In table 1 You present means and SDs for composite scores but You report that they significantly differ from normality (that is why You used MLR estimation). I suggest report quartiles instead,
3. Table 2: explanations of acronyms used in table header should be in notes below the table not in the title. And - it's not clear if fit indices obtained in subpopulations (i.e. males/females etc.) are in 1 factor, hierarchical or 3 factor model? Probably I'm not smart enough to find this information,
4. Table 5: I suggest reporting effect size for the differences. Or (if it's possible) maybe You could try to conduct MLR regression (even in SEM framework maybe? - i don't know possibilities of STATISTICA), this could give us the answer about the weights of this factors, probably more interesting than significance only,
5. Paragraph 3.7: Because of luck of normality I'm not sure if associations are linear - that is, correlations (I believe it's Pearson coefficient? I can't find any info if it's other) is not adequate if relation is not linear. I believe You should include some info about linearity or any data transformations done befor assessing these correlations.
6. Table 6 has some technical issues according to APA standards (i.e. strange horizontal borders). I suggests replace phrase: "Variable" in header on: "SAS-2" (if I'm right). In notes I suggest to explain names and acronyms used in the table (I know - they are described in methods, but I need to back few pages... But I'm lazy...)
7. Similarly I suggest notes in table 7.
8. Lines 354 - 360 You suggest that scale could be useful in detecting some 'psychiatric disorders' (if I understand Your intentions). I'm not sure - what do You mean? Are there any reports suggesting this kind of use in other populations? (I didn't find any).
Sum it up - please find my suggestions rather as technical, they do not change the overall high rating of the paper and Your work.
